# Overcoming Periodic Stripe Noise in Infrared Linear Array Images: The Fourier-Assisted Correlative Denoising Method

**DOI:** 10.3390/s23218716

**Published:** 2023-10-25

**Authors:** Weicong Chen, Bohan Li

**Affiliations:** 1Key Laboratory of Infrared System Detection and Imaging Technology, Chinese Academy of Sciences, Shanghai 200083, China; chenweicong@mail.sitp.ac.cn; 2Shanghai Institute of Technical Physics, Chinese Academy of Sciences, Shanghai 200083, China; 3University of Chinese Academy of Sciences, Beijing 100049, China

**Keywords:** infrared linear array image, low frequency, periodic noise, correlation-centric denoising

## Abstract

Infrared linear array detectors frequently experience vertical, low-frequency, and periodic stripe noise during imaging, stemming from electro-mechanical interference. Unlike conventional periodic disturbances, this interference showcases long periodicities and is uniquely columnar in orientation. Its presence, especially within the low-frequency domain, renders conventional filtering techniques ineffective and, at times, detrimental to image quality. Addressing this challenge, we introduce Fourier-Assisted Correlative Denoising (FACD), a correlation-centric denoising approach tailored for such unique interference patterns. This mechanism begins with the capture of a pure background image, inclusive of periodic noise, during the non-uniform correction phase of the infrared detector. Leveraging the noise’s frequency domain attributes, we extract a one-dimensional single-cycle noise signal. The infrared image is subsequently segmented into parts, and using the detected noise periodicity, the one-dimensional signals for each segment are computed. By leveraging the correlation between these signals and the benchmark one-dimensional noise pattern, we ascertain the noise profile within each segment. This profile is then employed for spatial domain denoising across the entire image frame. Empirical assessments confirm that the FACD outperforms contemporary denoising techniques by augmenting the peak signal-to-noise ratio by approximately 2.5 dB, underscoring its superior robustness. Furthermore, in light of its specificity to this noise model, FACD rapidly denoises high-resolution real infrared linear array scans, thus meeting the stringent real-time and resolution imperatives of advanced infrared linear array scanning apparatuses.

## 1. Introduction

As advancements continue in large-scale focal plane arrays and infrared linear array detector techniques [1,2], the 360-degree rotational scanning technique has increasingly caught the eye for its applicability to wide-angle imaging scenarios. Infrared linear array detectors have been effectively deployed in diverse fields, encompassing aerial target surveillance [3,4,5], medical diagnostics [6], facial recognition [7], industrial inspections [8], and aerospace applications [9]. Parallel to these developments, there has been a burgeoning interest in the research community toward refining denoising techniques for images obtained through infrared linear array detectors [10,11]. In imagery, disturbances related to infrared linear arrays primarily fall into stripe noises [12,13] and high-frequency mixing noises. The former can be further divided into those resulting from detector non-uniformities [14] and those arising from electromagnetic interferences with the detector [15].

Stripe noise, originating from detector inconsistencies, frequently demonstrates a directional preference, manifesting horizontally or vertically. This noise distribution does not adhere to a consistent pattern and typically corresponds to the detector’s photosensitive element’s response to illumination. Addressing the challenges posed by image stripes due to detector non-uniformity, Li Fangzhou et al. [11] proposed an enhanced NL-means model capable of concurrently eliminating stripe noise and Gaussian noise within images. In similar endeavors, Wang E et al. [16,17] amalgamated guided filtering with wavelet transformations, aiming to extract stripe noise from images within the frequency domain. Periodic stripe noise, attributed to electromagnetic disturbances, frequently presents with short intervals and high frequencies, while its direction generally does not appear horizontal or vertical. It is challenging to denoise this noise directly from the spatial domain. A prevailing method encompasses transitioning the image to the frequency domain, facilitating more efficient noise mitigation. In this vein, Hamd M H et al. [18] proposed a filter anchored in the noise signal’s peak value, designed to target and neutralize the periodic noise crest within the frequency domain. In a parallel approach, Yadav V P et al. [19] outlined a denoising methodology predicated on a localized threshold.

Infrared linear array detectors, when employed in 360-degree rotational scanning, require instantaneous imaging due to their rapid acquisition rates. A single pass can produce tens of thousands of image columns. This imaging modality introduces a peculiar kind of periodic stripe noise attributable to electromagnetic interference. Unlike noises from other devices, this noise showcases a protracted period and a lower frequency, aligning conspicuously with the direct current (DC) component within the frequency domain.

Notably, deploying denoising methodologies as suggested by Hamd M H et al. [18] and Yadav V P et al. [19], which primarily target peak noise in the frequency domain, might compromise the image’s low-frequency constituents, leading to marked image degradation. Additionally, the periodic noise in infrared linear array rotational scanning imagery is elongated, with gradual pixel transitions across adjacent columns, lacking any distinct singular stripes. As a result, denoising strategies, as posited by Li Fangzhou et al. [11] and Wang E et al. [16,17] encounter reduced efficacy. The enhanced NL-means paradigm presented by Li Fangzhou et al. [11] embeds a sliding search operation. Simultaneously, the methodologies proposed by Wang E et al. [16,17] intertwine guided filtering, the MHE algorithm, and TV-L1 regularization—factors that can potentially constrain the achievement of the real-time imaging prerequisites of infrared linear array systems. With the advancements in deep learning, denoising algorithms based on this framework have shown significant effectiveness [20,21]. However, these methodologies’ substantial dependency on curated image datasets is a significant limitation. The intrinsically high resolution of infrared linear array images complicates the task of compiling a comprehensive dataset, presenting formidable challenges. The self-supervised denoising method pioneered by Jaakko Lehtinen et al. [22], along with the approach by Qingliang Jiao et al. [23] designed to minimize reliance on pristine training data are particularly significant. While both methodologies cater to application requirements, the neural network model’s consistent input size necessitates size preprocessing for the THz spectrum, amplifying the complexity. Furthermore, the substantial computational demand of these techniques poses challenges for real-time processing of high-resolution images.

Considering the abovementioned challenges, this paper proposes an infrared linear array image denoising algorithm based on signal correlation. The foundational framework of the denoising algorithm hinges on a correlation-centric approach. Specifically, our proposed algorithm, FACD (Fourier-Assisted Correlative Denoising), operates by leveraging the intrinsic frequency domain attributes of periodic noise. At its core, FACD begins by capturing a background image saturated with the said noise. It then delves into the frequency spectrum to extract a one-dimensional, single-cycle noise signal. By segmenting the infrared image and matching the detected noise periodicity, the algorithm harnesses the correlation between these segmented signals and the established one-dimensional noise pattern. This correlation-driven methodology enables FACD to precisely ascertain the noise profile within each segment of the image, facilitating effective spatial domain denoising. The primary advantage of the FACD approach lies in its precision and specificity to target the unique interference patterns presented by electromechanical interferences in infrared linear array detectors. Experimental results show this algorithm proficiently eliminates periodic noise in high-resolution 360-degree rotational scanning images, satisfying the real-time imperatives of infrared linear array detectors.

## 2. Proposed Methods

### 2.1. Characteristics of the Periodic Noise

Periodic noise is a prevalent noise type in images, typically emanating from interference during image capture or subsequent transmission. Traditionally, this noise is typified by its high frequency, manifesting as luminous sectors within the image’s two-dimensional Fourier transform domain, as delineated in Figure 1. An example would be a newspaper image sampled under the Moiré pattern.

This paper concentrates on a distinct type of noise, divergent from traditional periodic disturbances. Infrared linear array detectors capture the images under scrutiny during their comprehensive 360-degree rotational scanning. Electromechanical disturbances during this procedure introduce noise with a lower frequency. In the spatial domain of the image, this noise manifests as subtle, gradually varying stripes. Conversely, within the image’s two-dimensional Fourier transform domain, it is observed as luminous sectors proximate to the DC component. The noise image of the infrared detector obtained through blackbody is shown in Figure 2a, and its two-dimensional Fourier frequency domain image is presented in Figure 2b.

Upon juxtaposing the frequency domain images of periodic noise depicted in Figure 1 and Figure 2, one discerns a clear contrast. Traditional periodic noise, when viewed in the frequency domain, is characterized by multiple bright regions distanced from the DC component. It indicates that such noise primarily comprises high-intensity, high-frequency components, significantly affecting image quality. Conversely, the infrared linear array frequency domain image scarcely showcases distinct luminous areas, revealing only sporadic bright spots proximate to the DC component. It underscores the predominance of noise characterized by reduced intensity and frequency. Employing a standard Gaussian notch filter to address this periodic noise can unintentionally weaken the image’s inherent low-frequency elements, negatively impacting the denoising outcome. To better address the low-frequency periodic noise for denoising, analyzing the infrared noise image and introducing some prior information is essential. Examining the pure background noise image sourced from the blackbody, it is evident that the noise primarily appears as vertical stripes with varying brightness. Consequently, a comprehensive statistical analysis of the noise is conducted in both horizontal and vertical orientations, as delineated in Figure 3.

As illustrated in Figure 3, the noise distinctly displays a periodic characteristic along the image’s columnar axis, whereas along the row axis, it resembles stochastic Gaussian noise. Comprehensive assessments using the blackbody substantiate that the amplitude of this periodic noise remains unaffected by variations in the blackbody’s temperature. This inference posits that the periodic noise prevalent in infrared detectors originates exclusively from electromechanical interference, devoid of influence from other external variables. Every time the infrared linear array scanner is powered on, it necessitates a non-uniformity adjustment, obtaining the images of periodic noise under a pure background. Subsequently, the noise period could be determined, and denoising could be performed in the spatial domain based on signal correlation.

### 2.2. Noise Period Calculation Based on Fourier Transform

Normally, infrared detectors need to be calibrated to ensure consistent response from the detection unit before each use. In this study, the non-uniformity correction deployed for the infrared detector is based on the single-point method. Given that the infrared images acquired during this correction phase primarily reflect the pure background, they remain relatively insulated from external scene perturbations. Such images exhibit a pronounced presence of periodic noise. Therefore, the algorithm employs the Fourier transform to analyze the images obtained during non-uniformity correction, aiming to discern the noise periodicity attributable to electromechanical interference. In this example, upon examination of the periodic noise within the pure background imagery, it was ascertained that the noise periodicity for the selected infrared detector approximates 850 columns. Furthermore, recognizing the high resolution inherent to infrared linear array scanning images and the need for multiple noise periods in Fourier transform calculations, this section crops the line scanning images, adjusting their resolution to 1024 × 4096.

Since the periodic noise of infrared linear array images primarily shows in the column direction, the clipped image is first converted into a one-dimensional signal in the column direction, as shown by the following formula:(1)$$I_{c1}\left(i\right)=\frac{1}{N_{row}}\times \sum_{j=1}^{N_{row}}I_{c}(i,j)$$
where $I_{c}$ is the image used for non-uniform correction, i, the number of columns, obtains $i\in \left[1,N_{col}\right]$, where $N_{col}=4096$; $N_{row}=1024$ is the number of rows of the image. Finally, $I_{c1}$ is the transformed one-dimensional signal sequence, as shown in Figure 3a.

In the next step, Fourier transform is applied to $I_{c}$, as shown in the following formula:(2)$$X\left(m\right)=\sum_{i=1}^{N_{col}}I_{c1}\left(i\right)\cdot e^{-j\frac{2\pi \left(i-1\right)m}{N_{col}}}$$

In this formula, $X\left(m\right)$ is the first half of the sequence after the Fourier transform, excluding the DC component, where $m\in \left[1,\frac{1}{2}N_{col}\right]$. Using the modulus values of $X\left(m\right)$, the maximum value of $X\left(m\right)$ is calculated, then record the current subscript $m_{MAX}$ is recorded, as shown in the following equation:(3)$$abs\left(X\left(m_{MAX}\right)\right)=\max\left(abs\left(X\left(m\right)\right)\right)$$

In this context, the function max(·) retrieves the maximum value, while abs(·) performs the modulus operation on complex numbers. The noise period’s estimated value, denoted as $P_{e}$, is elucidated in the subsequent equation:(4)$$P_{e}=\frac{N_{col}}{m_{MAX}}$$

Since the number of columns of the cropped image $N_{col}$ may not necessarily be an integer multiple of the noise period, $P_{e}$ is the estimated value of the noise signal period. The closer the number of columns of the cropped image is to an integer multiple of the noise period and the larger that number, the closer $P_{e}$ is to the actual noise period.

In addition, other noises that are present in the collected signal can also influence the estimated value $P_{e}$ of the electromechanical noise period. Using $P_{e}$ as a constraining parameter and leveraging the periodicity of $I_{c1}$, our objective is to accurately determine the noise period, denoted $\hat{P}$.

Initially, the median value, ‘Mid’, of the sequence $I_{c1}$ is calculated as indicated in the formula below:(5)$$Mid=\frac{1}{2}\times \left(\max\left(I_{c1}\right)+\min\left(I_{c1}\right)\right)$$

In this context, min(·) represents the function used to determine the minimum value, while max(·) signifies the function to ascertain the maximum value. Subsequently, we find points in the sequence $I_{c1}$ that are near ‘Mid’, as demonstrated in the following equation:(6)$$P_{array}=Find\left(Mid-V_{1}\leq I_{c1}\leq Mid+V_{1}\right)$$

Within this framework, $P_{array}$ denotes a set of points in $I_{c1}$ proximate to the median ‘Mid’. The function Find(·) is used to identify values in $I_{c1}$ that satisfy designated conditions. In this equation, $V_{1}$ acts as a range threshold parameter. Given that the infrared image of this device is articulated in 8-bit grayscale, this threshold is pragmatically set at 2. The objective is to discern values in the $I_{c1}$ series that fluctuate between the $\left[Mid-2,Mid+2\right]$ range. With the noise period estimate serving as a benchmark restraint, the precise value $\hat{P}$’s computation is illustrated in the subsequent equation:(7)$$T_{max}=mean\left(P_{array}\left(Find\left(P_{array}>P_{array}\left(1\right)+P_{e}-V_{2}\;\&\;P_{array}<P_{array}\left(1\right)+P_{e}+V_{2}\right)\right)\right)$$
(8)$$T_{min}=mean\left(P_{array}\left(Find\left(P_{array}>P_{array}\left(1\right)-V_{2}\;\&\;P_{array}<P_{array}\left(1\right)+V_{2}\right)\right)\right)$$
(9)$$\hat{P}=T_{max}-T_{min}$$

In this discussion, $\hat{P}$ represents the definitive noise period value, while $P_{array}\left(1\right)$ extracts the first value from the $P_{array}$ series. The mean(·) function procures the average value, and the Find(·) function is employed to pinpoint values within $P_{array}$ that resonate with the pre-established criteria. As evident from Figure 3, which delineates the noise image’s columnar orientation statistical depiction, the electromechanical noise periodicity is roughly around 800 columns. The range threshold $V_{2}$ in the formula is manually set to 25 with the intention to constrain the range for $\hat{P}$ so that $\hat{P}\in \left[700,900\right]$, ensuring its proximity to the actual noise period.

### 2.3. Fourier-Assisted Correlative Denoising (FACD) Mechanism

The periodic noise in the infrared columnar image caused by electromechanical interference is a type of additive noise, and its image signal model is shown in Equation (10).
(10)$$v_{i}=u_{i}+s_{i}+t_{i}$$

In this context, $v_{i}$ represents the original infrared imagery filled with noise disruptions, while $u_{i}$ signifies the ideal noise-free infrared imagery. $s_{i}$ delineates the periodic noise due to electromechanical disturbances, and $t_{i}$ captures other noise modalities. The aim of the algorithm in this subsection is to utilize the correlation of periodic signals to eliminate the periodic additive noise on the infrared image. As observed from Section 2.2, even with our computational derivation of the noise image’s periodic magnitude $\hat{P}$ via Fourier transformation, direct denoising of this noise image remains challenging. This challenge arises when the linear array detector acquires images in a scanning fashion. Even if multiple images represent the same scene, the moments of capture might vary, leading to potential differences in the phase offset of noise stripes, as depicted in Figure 4.

Based on the analysis of the linear array detector’s periodic noise in Section 2.1, the periodic noise possesses a robust one-dimensional periodic characteristic in the column direction. It is feasible to exploit the correlation in the noise signal’s columnar direction to determine the phase of the periodic noise in the image. The formulas for auto-correlation and cross-correlation of the signal are provided in the succeeding equations.
(11)$$R_{xx}\left(l\right)=\frac{1}{N}\sum_{n=1}^{N}x\left(n\right)x\left(n-l-\min\left(sign\left(n-l\right),0\right)\times N\right)$$
(12)$$R_{xy}\left(l\right)=\frac{1}{N}\sum_{n=1}^{N-1}x\left(n\right)y\left(n-l-\min\left(sign\left(n-l\right),0\right)\times N\right)$$

Let $x\left(n\right)$ and $y\left(n\right)$ be one-dimensional sequence signals with $\in \left[1,N\right]$, where N signifies the length of the sequence and sign(·) is the signum function. During the non-uniformity correction stage of the detector, a periodic noise image is secured against a pure background, as exemplified in Figure 2a. The average value for each column of the image is determined as outlined in Equation (13).
(13)$$n(j)=\frac{1}{R}\sum_{i=1}^{R}I_{n}(i,j)$$

In this context, n(j) refers to the one-dimensional sequence depicted in Figure 3a, which is recognized as the multi-cycle noise present in the image’s columnar orientation. The term $I_{n}(i,j)$ signifies a pixel within the noise image when set against a pure background. Here, with $\in \left[1,C\right]$, $i\in \left[1,R\right]$, where C represents the total number of columns in the image and *R* indicates its rows. From the sequence n(j), we extract a contiguous sequence with length paralleling the noise cycle $\hat{P}$, termed n(j), where j is within the interval $j\in \left[1,\hat{P}\right]$.

To address the infrared image affected by periodic noise, it is segmented into individual blocks. Each block has a column size matching the noise cycle $\hat{P}$ and row size approximating 1/4 of the complete frame’s row count. This modular decomposition is graphically represented in Figure 5.

The subsequent phase involves computing the one-dimensional sequence in the column direction for each subdivided block, as represented by the following equation:(14)$${I^{\prime }}_{p}\left(j\right)=\frac{4}{R}\sum_{i=n\times \frac{R}{4}+1}^{\left(n+1\right)\times \frac{R}{4}}I\left(i,j+m\times \hat{P}\right)$$

In this equation:R denotes the row count of the image $I\left(i,j\right)$;n stands for the count of segmentation blocks in the row direction and $n\in \left[0,3\right]$;m stands for the count of segmentation blocks in the column direction; $m\in [0,floor(C/\hat{P})-1]$, where the function floor(·) rounds a number down to its nearest integer, C signifies the column count of the image.

Use Formula (14) to calculate the offset ${m\prime }_{p}$ at which the cross-correlation between each small block’s one-dimensional column direction sequence ${I^{\prime }}_{p}(j)$ and n′(j) reaches its maximum. At this time, the length of the sequence $N=\hat{P}$, and ${m\prime }_{p}$ is transformed as shown in the following formula:(15)$${m^{\prime }}_{p}=\left\{\begin{array}{c} {m^{\prime }}_{p},\left|{m^{\prime }}_{p}\right|<\left|\hat{P}-{m^{\prime }}_{p}\right|\\ {m^{\prime }}_{p}-\hat{P},others \end{array}\right.$$

Sort all the ${m^{\prime }}_{p}$ values to obtain the sequence ${m^{\prime }}_{p}\left(i\right),i\in \left[1,N_{m}\right]$. Here, $N_{m}$ is the number of small image blocks as shown in Figure 5. Take the middle portion of the sorted ${m^{\prime }}_{p}$ sequence offsets and compute their mean to obtain the corresponding one-dimensional noise sequence for the small image block, as shown in the following formula:(16)$$n_{p}(k)=\left\{\begin{array}{c} n\left(k-mean\left({m^{\prime }}_{p}\right)+\hat{P}\right),k-mean\left({m^{\prime }}_{p}\right)<0\\ n\left(k-mean\left({m^{\prime }}_{p}\right)-\hat{P}\right),k-mean\left({m^{\prime }}_{p}\right)>\hat{P}\\ n\left(k-mean\left({m^{\prime }}_{p}\right)\right),others \end{array}\right.$$
where $mean\left(\cdot \right)$ is the average function and $k\in \left[1,\hat{P}\right]$. Use $n_{p}(k)$ to denoise the image $I\left(i,j\right)$, as shown in the equation:(17)$$I_{u}\left(i,j\right)=\left\{\begin{array}{c} I\left(i,j\right)-n_{p}\left(mod\left(j,\hat{P}\right)\right),mod(j,\hat{P})\neq 0\\ I\left(i,j\right)-n_{p}\left(\hat{P}\right),mod\left(j,\hat{P}\right)=0 \end{array}\right.$$
where $I_{u}\left(i,j\right)$ is the image after denoising using correlation, mod(·) is the modulo function, and $\hat{P}$ is the noise period obtained using the Fourier Transform in Section 2.2.

### 2.4. Overall Algorithmic Procedure

As illustrated in Figure 6, the comprehensive algorithmic procedure is delineated systematically as follows:Step 1: Single-Cycle Noise Sequence Extraction $n\prime \left(j\right)$During the non-uniformity correction process of the infrared camera, collect the cyclical noise image under a pure background;Implementing the Fourier Transform-centric periodic solution algorithm, as introduced in Section 2.2, to identify the noise period $\hat{P}$ for the observed noise signal;Employing Formula (13) and the ascertained noise period $\hat{P}$, derive the single-cycle noise sequence $n\prime \left(j\right)$.Step 2: Image Data Partition and Processing

Begin by subdividing the procured images into specific blocks, as illustrated in Figure 5. Subsequently, for each image block, compute the one-dimensional sequence ${I\prime }_{p}\left(j\right)$ utilizing Formula (14).


Step 3: Cross-Correlation Function and Offset Determination


For each one-dimensional image sequence ${I\prime }_{p}\left(j\right)$, its cross-correlation function with the noise sequence $n\prime \left(j\right)$ is calculated using Formula (12). Record the offset ${m\prime }_{p}$ when each cross-correlation function reaches its maximum value. Thereafter, a transformative Equation (15) is applied to the offset ${m\prime }_{p}$.


Step 4: Sequence Ordering and Offset Refinement


After the previous steps, the offset series ${m\prime }_{p}$ is systematically arranged to form the sequence ${m^{\prime }}_{p}\left(i\right),i\in \left[1,N_{m}\right]$. Herein, $N_{m}$ represents the total number of image blocks, as explicated in Figure 5. Eliminate both the top quartile of the larger sequence values and the bottom quartile of the smaller ones. Following this, Equation (16) is employed to produce the one-dimensional noise sequence $n_{p}\left(j\right)$, constrained by $j\in \left[1,\hat{P}\right]$.


Step 5: Image Denoising


Collaboratively, with the sequence $n_{p}\left(j\right)$, Equation (17) is leveraged to execute a denoising process on the image.

## 3. Experimental Results and Analysis

### 3.1. Investigation into the Consistency of Periodic Noise in Infrared Linear Array Detectors

The preceding section articulated that the Fourier-Assisted Correlative Denoising (FACD) mechanism requires an image characterized by periodic noise against a pure background. We conducted a comprehensive experiment to assess the applicability and robustness of the proposed mechanism in real-world scenarios. For the empirical study, the experiment was designed to meticulously record and analyze the fluctuations in noise periodicity and amplitude over successive time intervals following the initial power-up of the detector without undergoing any rebooting processes.

We employed a 1024-pixel long-wave infrared linear array detector, which boasts a specialized shutter mechanism. This shutter is purposefully activated when a non-uniformity correction command is initiated, facilitating the precise non-uniformity correction procedure. The images obtained at this critical juncture form the foundation of our data source. By applying histogram statistical methods, we discerned the columnar distribution associated with the noise periodicity within the current image framework. Data were harvested across varied operational intervals, culminating in four distinct datasets. Each of these sets encapsulated an average of 55 noise cycles. The resultant histograms are illustrated in the subsequent figure. Within this graphical representation, the x-axis demarcates the column count corresponding to the noise periodicity, while the y-axis quantifies the cycle counts spanning diverse noise periods.

Figure 7 provides a sequence of histograms documenting the performance of the infrared detector at different intervals. Specifically, Figure 7a showcases the data shortly after the initial power-on and subsequent cooling. The following histograms, Figure 7b–d, were captured at 6 h intervals, marking 6, 12, and 18 h of continuous operation. A consistent observation across these figures is the stable noise periodicity, consistently measuring 847 columns. 

For a deeper insight into noise amplitude fluctuations, we compiled and analyzed data from 220 noise periods across the four intervals. Figure 8 presents this analysis, where the x-axis demarcates experimental batches of 22 periodic signals and the y-axis displays amplitude variations in pixel values. The noise amplitude across all periods demonstrated remarkable stability, fluctuating minimally between −30 and 30 pixels.

Drawing from these observations, the infrared linear array detector for the experiment, once powered up, maintains consistent noise characteristics in terms of period and amplitude. This consistency underscores the effectiveness of the FACD algorithm. Initialized with a single-period noise signal extracted at power-up, the FACD algorithm can consistently denoise images without needing recalibration during extended operation periods.

### 3.2. Impact of Image Block Width

As depicted in Figure 5, this approach employs a strategy of partitioning the image into blocks to enhance the denoising results of the correlation-based algorithm. The length of each image block corresponds to the signal period deduced from the algorithm in Section 2.2, while the width is manually set to a quarter of the entire frame’s row count. During real-world capture by the camera, the impact of scene elements within the image on inter-signal correlation cannot be disregarded. When the width of an image block is set smaller, more blocks can be derived from an infrared linear array image of the exact resolution. Since fewer pixels exist within a smaller block, the calculation time for the one-dimensional column-direction signal of the block image is reduced. However, the signal within the block is largely determined by the scene elements, making the influence of the image scene on the correlation algorithm more prominent.

Conversely, when one opts for larger block widths, the number of blocks extracted from an infrared linear array image of the exact dimensions is reduced. The calculation time for the column-direction signal increases. Given that the correlation between the image scene and periodic noise is relatively weak, the influence of the image scene on the correlation algorithm is considerably reduced when the block’s width is larger.

In this subsection, we conduct an experiment utilizing 20 infrared line-scanning images, each with a resolution of 1024 × 4096. A pure background periodic noise image of identical resolution is acquired through a blackbody. We create a simulated infrared image with noise by superimposing this periodic noise image onto the clean infrared line-scanning image. Using the algorithm described in Section 2.2 and the pure background noise image, we identify a signal period covering 847 columns. From these data, we derive a single-cycle, one-dimensional noise signal sequence. For the batch of 20 noisy infrared images, we randomly pick image segments sized 1024 × 847 and segment them further into blocks. The subsequent denoising process is executed using the approach described in Section 3.2. The relationship between the block width of the images and their corresponding denoising results is illustrated in Figure 9.

Figure 9 displays the average PSNR value across 20 image frames. With an image block width of 1, every row of a 1024 × 847 resolution image is correlated with a singular noise cycle. Even though there are 1024 such image blocks, the scene details notably affect signal correlation, resulting in a post-denoising PSNR average of 30.95 dB. As the image block width expands to four, the number of smaller blocks within the 1024 × 847 image diminishes. However, the scene details’ impact on signal correlation decreases, holding the denoised image’s PSNR steady at around 32 dB, indicating improved denoising. Given that the infrared images used in this study have 1024 rows, for simplification, further experiments segmented the authentic images into blocks, each sized 256 × 847.

### 3.3. Robustness Experiment

In the context of image acquisition with infrared linear array detectors, the content of the captured scenes significantly influences the denoising effectiveness of the algorithm presented in this paper. For images that primarily feature background elements, the fluctuations in the one-dimensional signal, as elaborated in Section 2.3, are primarily dictated by inherent periodic noise. Thus, the algorithm leverages signal correlation to achieve denoising results. However, when the captured image encompasses intricate background details, the changes in the one-dimensional signal will culminate in both the scene elements and the periodic noise. Given the unpredictable and variable nature of scene content, it can be inferred that such content operates independently from the periodic noise signal.

To systematically replicate the interplay of scene content on the denoising efficiency of the algorithm, this section incorporates a one-dimensional signal characterized by a Gaussian distribution centered at zero and exhibiting diverse variances. This signal is superimposed onto the one-dimensional signal permeated with periodic noise, computed per the methodology above. The underlying logic remains lucid: an augmentation in the variance of the Gaussian distribution signifies increased scene complexity, as illustratively presented in Figure 10.

In Figure 10a, the depicted curve elucidates a one-dimensional signal derived from a chosen image block extracted from an infrared linear array image infused with periodic noise. This periodic noise-pervaded image was synthesized by overlaying a noise-free image with a pure background noise image.

Figure 10b through Figure 10h, respectively, display the outcomes after adding one-dimensional Gaussian noise sequences with zero mean and varying variances to the signal in Figure 10a. In this context, σ denotes the standard deviation of the Gaussian noise.

To verify the influence of the one-dimensional Gaussian noise on the denoising efficacy of the presented algorithm, an experimental setup, encapsulated within this section, leveraged 20 frames of immaculate real infrared linear array rotational scanning images, each boasting a resolution of 1024 × 4096. By overlaying these images with the pure background periodic noise sourced from the blackbody, we achieved 20 frames of infrared linear array images replete with periodic noise and identical resolution parameters. 

As the denoising mechanism of our algorithm took course, diverse variances of Gaussian noise sequences were added to the one-dimensional signal extracted from the periodic noise-laden linear array images. Table 1 lucidly tabulates the aggregate outcomes, underscoring the influence of the one-dimensional Gaussian noise on the algorithm’s efficacy.

In the results presented in Table 1, it is evident that even with an increase in the standard deviation, σ, of the overlaid one-dimensional Gaussian noise continuously increasing, the PSNR (peak signal-to-noise ratio) of the denoised image remains relatively stable. It underscores the robustness of the proposed algorithm.

### 3.4. Simulation Experiment

To substantiate the denoising efficacy of the presented algorithm, we undertook simulation experiments utilizing noise-free infrared images. We sourced 50 infrared frames from the CAMEL dataset [24] and incorporated one pure background infrared linear array image, exhibiting periodic noise, captured via a blackbody. This noise image, with a resolution of 1024 × 4096, exhibits a noise periodicity spanning roughly 850 columns. Given that the CAMEL dataset [24] contains infrared images of 480 × 640 resolution and acknowledges the additive characteristic of infrared linear array periodic noise, it was imperative to integrate this noise into our selected dataset. The infrared images were resized to 1024 × 1365 from their original 480 × 640 by employing interpolation techniques. The columnar dimension of the noise image was then constricted to produce a resolution of 1024 × 1365 and an inferred noise periodicity of about 284 columns. By computing the mean of the noise image, each pixel value was reduced by this mean and then added to the infrared set, forming a simulated dataset with additive noise.

We then applied two periodic noise removal algorithms, Two-Circle Filter [18] and Local Threshold [19], and three improved infrared image stripe noise denoising methods: the enhanced NL-means algorithm [11], the Wavelet Gradient-Equalization algorithm [16], and the Wavelet TV-L1 algorithm [17]. Due to the noise’s low-frequency characteristics, the Two-Circle Filter [18] was adjusted to circumvent the image’s DC component by configuring the filter radius d to 2. 

The enhanced NL-means algorithm [11] operated with neighborhood search blocks sized at 3 × 3, governed by a parameter γ=500. Both the Wavelet Gradient-Equalization [16] and Wavelet TV-L1 [17] algorithms employed the Haar wavelet basis with a horizontal guiding filter window of 1 × 8 and a vertical one of 10 × 1, and a guided filter parameter ε=0.16. The Wavelet Gradient-Equalization algorithm [16] set the MHE algorithm parameter n to 2, indicating that it processes two adjacent columns of the pixel. In the proposed method, 16 image blocks were selected in Step 2. Five images were handpicked from the dataset to qualitatively analyze the denoising effects of different algorithms, as shown in Figure 11. 

As discerned from Figure 11, it is evident that our algorithm provides the best denoising effect. The denoised image is virtually indistinguishable from the noise-free image, devoid of discernible noise stripes. However, denoised images obtained from the Two-Circle Filter [18] and Local Threshold [19] still exhibit periodic noise stripes. The low frequency of infrared linear array periodic noise primarily affects the image’s low-frequency components near the DC, unlike other high-frequency periodic noises, which manifest clear peak noise frequencies in their spectra. Solely suppressing noise peak frequencies in the spectrum is not enough for effective denoising. Due to the extended noise period and minimal changes in adjacent column pixels, there are no noticeable stripe noises, making the enhanced NL-means [11], Wavelet Gradient-Equalization [16], and Wavelet TV-L1 [17] less effective. Our proposed strategy integrates a priori knowledge of the intrinsic periodic noise in the background. Through meticulous analysis of the periodic attributes of the noise, our algorithm studies the correlation between noise patterns and image content and then denoises images in the spatial domain. The results are superior. For a quantitative assessment, the pristine noise-free image served as the benchmark against which the PSNR of the denoised image was ascertained, with the detailed outcomes delineated in Table 2, followed by a line chart (Figure 12) to further present the trends.

Table 2 evaluates the average PSNR across the 50 denoised infrared frames. Our method outperforms other algorithms. The PSNR of images denoised using the Two-Circle algorithm [18] was the lowest. It can be attributed to the proximity of low-frequency periodic noise peaks in the frequency domain to the DC component. 

Despite calibrating the filter radius r to a value of 2, the intrinsic operations of the denoising algorithm might have inadvertently perturbed the DC component within the frequency domain, consequently altering the global mean value of the image frame. Such a phenomenon correlates with the visually diminished luminosity observed in images processed using the Two-Circle algorithm, as depicted in Figure 11. 

Table 3 delineates the various denoising algorithms’ SSIM (structural similarity) indexes. It is evident that our proposed method consistently achieves superior results in preserving structural integrity. While the Wavelet TV-L1 approach and other traditional algorithms deliver commendable performance, the SSIM index for our proposed method is consistently closer to unity, underscoring its ability to retain the inherent structure of the image post-denoising. This superior structural preservation, as highlighted in our algorithm, complements the PSNR results presented earlier and further consolidates the efficacy of our method, as vividly depicted in the comparison of the datasets.

### 3.5. Real Data Experiment

To validate the effectiveness of the proposed algorithm in the context of authentic high-resolution infrared linear array images, this section employs two real infrared linear array scanned images with dimensions of 1024 × 4096 containing periodic noise for the denoising experiment, along with one pure background noise image obtained during the equipment’s non-uniformity correction process. For the Wavelet Gradient-Equalization algorithm [16] and the Wavelet TV-L1 algorithm [17], the vertical guiding filter window dimensions were modified to 100 × 1, optimizing their capacity to extract stripe noise in high-resolution imagery. Other algorithm parameters remain consistent with those in Section 3.4. The denoising results are graphically represented in Figure 13. 

In Figure 13, the proposed algorithm effectively removes periodic noise while preserving the details of the image, especially the intricate features within the scene. On the other hand, the Two-Circle Filter [18] and Local Threshold [19] algorithms significantly blur the detailed information about the clouds in the sky. Such distortion can be traced back to the confluence of low-frequency elements, such as the clouds mentioned above, and the frequency domain characteristics of the periodic noise. This overlap poses a distinct challenge to denoising endeavors, mainly when conducted directly within the frequency domain, further compounded by increasing image resolution.

Furthermore, in the real infrared linear array-scanned images, the noise period is approximately 850 columns, much higher than that in the simulated images. Consequently, the change in pixel values between adjacent image columns is even more gradual. Such slow variation exacerbates the inadequacy of the improved NL-means algorithm [11], Wavelet Gradient-Equalization algorithm [16], and Wavelet TV-L1 algorithm [17] when confronted with this specific type of noise. Their denoising efficacy under these circumstances is less than ideal.

### 3.6. Real-Time Performance of the Algorithm

Given the fast imaging capabilities and high resolution provided by infrared linear array detectors, it is imperative for associated algorithms to maintain optimal real-time performance. In this section, we assess the denoising time for each algorithm, utilizing authentic infrared linear array images (resolution 1024 × 4096) afflicted with periodic noise. The specifics of these algorithms remain consistent with Section 3.5. The computational environment comprises an Intel i7-9750H processor with 16 GB of memory, operating on the Matlab2020b platform.

Table 4 delineates the average runtime for each algorithm to denoise an individual image frame across ten trials. Notably, the enhanced NL-means algorithm [11] demands computational time for proximity search around every pixel, thereby prolonging its duration. The Wavelet Gradient-Equalization [16] and Wavelet TV-L1 [17] algorithms necessitate MHE and L1 regularization processes post-wavelet decomposition on sub-images (resolution 512 × 2048). This requirement incrementally extends their processing durations, making them less compatible with the real-time demands of infrared linear array detectors. Contrastingly, our proposed algorithm capitalizes on the unique periodic noise variation in the columnar direction. By leveraging periodic signal correlation, it facilitates direct spatial domain image denoising, ensuring minimal time consumption.

As the image resolution escalates, there is an evident proportional increase in the runtime across all algorithms. This increment is particularly pronounced in methods like the improved NL-means, which exhibit a substantial time surge even with modest increments in resolution. However, a compelling observation is the minimal runtime escalation of our proposed FACD method, which manifests only slight increments even when the image resolution multiplies. This indicates the scalability and robustness of our approach, making it particularly adept for applications that might necessitate varied image resolutions. Furthermore, when juxtaposed with other algorithms, the FACD’s time increment remains negligible, reinforcing its prowess in ensuring consistent and rapid denoising irrespective of resolution expansions.

Our algorithm emerges as a potent tool within infrared linear array imaging systems. According to the characteristics of this system, it first needs to close the baffle to perform the non-uniformity correction every time it powers up. Collect a frame of the image as the pure background image at this moment. Subsequent analysis yields noise periodicity and amplitude. The baffle is then reopened, and leveraging our algorithm’s methodology, the noise’s correlation with authentic images is ascertained, facilitating spatial domain denoising. After completing the above steps, the image denoising process is simply arithmetic operations on every pixel after receiving a column image. This streamlined approach ensures brevity in execution, addressing long-standing challenges posed by periodic noise due to electromechanical disturbances within infrared linear array imaging systems.

## 4. Discussion

Within the confines of the experiments outlined in the preceding section, our proposed algorithm exhibited marked proficiency in attenuating low-frequency noise, while simultaneously upholding commendable real-time operational efficiency. When one broaches the subject of high-resolution, real-time image processing systems, it becomes pivotal to note that the periodicity of columns in infrared linear array detectors might shrink to approximately 30 microseconds. Such systems predominantly find utility in the domains of target extraction and recognition. Given the prodigious generation rate, surpassing 30,000 image columns every second, resorting to a full-frame buffering approach for image elaboration would entail a substantial memory overhead, jeopardizing the real-time assurance. Conventionally, systems of this ilk adopt a staggered processing paradigm, buffering at intervals of 128 or 256 columns. In other words, with every caching instance, the system extracts targets within the image to achieve real-time target output. Thus, the advantages of column-by-column noise reduction are evident. Our algorithm can be deployed during the embedded image preprocessing phase. After performing rapid noise reduction on each column, the image is passed to the next stage. The resultant computational footprint is minimal, both in terms of resource consumption and parameter storage overhead. Such efficiency earmarks ample computational latitude for downstream algorithms, rendering our method an ideal fit for embedded systems that are relatively resource-constrained. 

However, our algorithm requires a preliminary step—namely, collecting pure noise images against a blackbody or a baffle. It may not resonate well with imaging systems without a baffle or a blackbody. Although one could obtain relatively clean background images by tilting the detector toward the sky, this method has limitations (for instance, when the working environment is indoors or when the sky has complex cloud layers). Future endeavors could contemplate refining this preliminary step, striving for simultaneous acquisition of periodic noise frequencies while not compromising the real-time operation. While there is potential to dovetail neural network frameworks into our model, a salient point of contention remains. As underscored earlier, extant deep convolutional neural networks (CNNs) tailored for noisy image mitigation predominantly fixate on entire frame denoising. Their deployment complexities, especially when juxtaposed with real-time processing demands, stand as impediments. Additionally, the conspicuous absence of high-resolution training datasets further hinders the optimization and effectiveness of such neural networks.

## 5. Conclusions

This research offers a novel solution to the prevalent challenge of low-frequency periodic signal noise induced by electromechanical interferences in infrared linear array detectors. We introduced an efficient denoising algorithm grounded in signal correlation principles, focusing on extracting and utilizing noise signals from pure backgrounds to achieve comprehensive denoising. The robustness and efficiency of our approach were validated through extensive simulations and real imagery tests, demonstrating notable enhancements in PSNR and superior denoising capabilities compared to existing methods. Notably, our algorithm maintained high performance even under substantial perturbations and noise variations, evidencing its adaptability to different noise conditions. Furthermore, the algorithm exhibited remarkable real-time efficiency in processing high-resolution images, showcasing its practical applicability and readiness for deployment in the embedded image preprocessing phase, ensuring minimal computational footprint and resource consumption. However, it is crucial to note that our algorithm necessitates a preliminary step of collecting pure noise images, which may pose limitations in certain working environments or system configurations. Future endeavors could focus on refining this preliminary step and exploring the integration of neural network frameworks, addressing the deployment complexities associated with real-time processing and the lack of high-resolution training datasets.

## Figures and Tables

**Figure 1 sensors-23-08716-f001:**
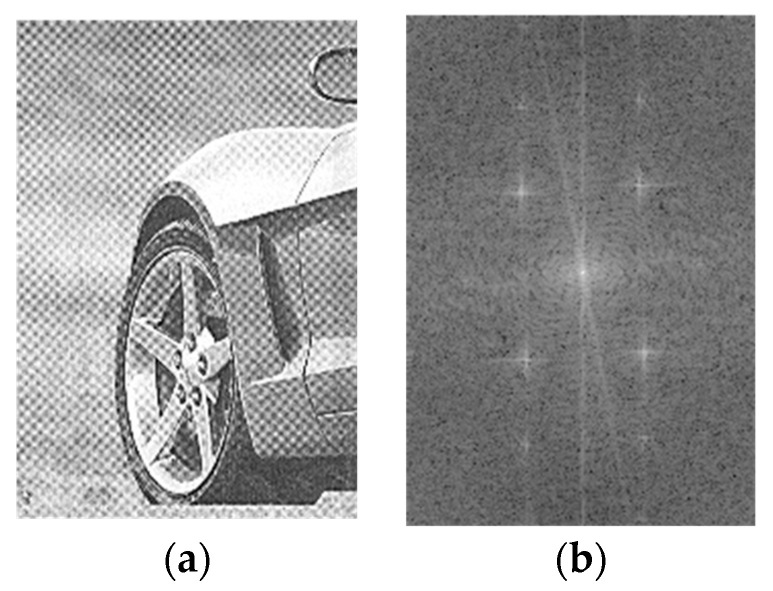
(**a**) Traditional periodic noise. (**b**) The spectrum of the traditional periodic noise.

**Figure 2 sensors-23-08716-f002:**
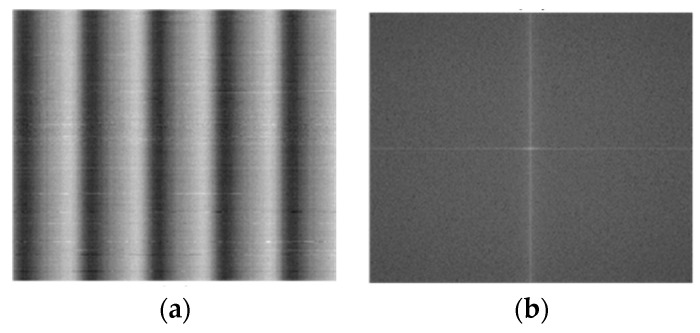
(**a**) Noise image obtained through blackbody. (**b**) Two-dimensional Fourier frequency domain image.

**Figure 3 sensors-23-08716-f003:**
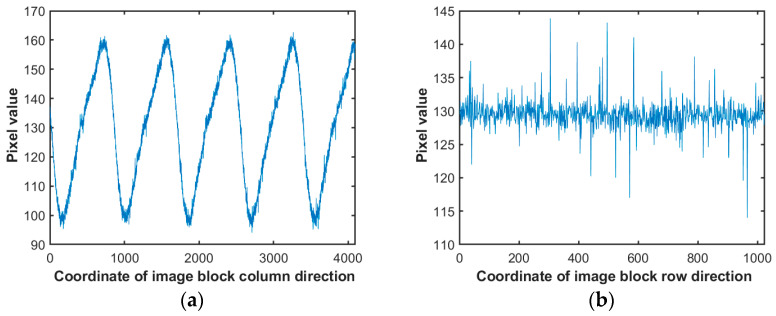
Analysis of infrared linear array scanning images: (**a**) Statistics of pixel mean values for each column of images. (**b**) Statistics of pixel mean values for each row of images.

**Figure 4 sensors-23-08716-f004:**
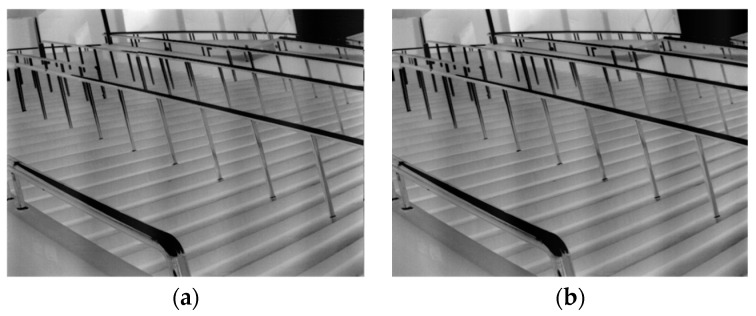
The same scene contains noise of different phases. (**a**) The image with noise in one phase (**b**) The image with the same noise but different phase.

**Figure 5 sensors-23-08716-f005:**
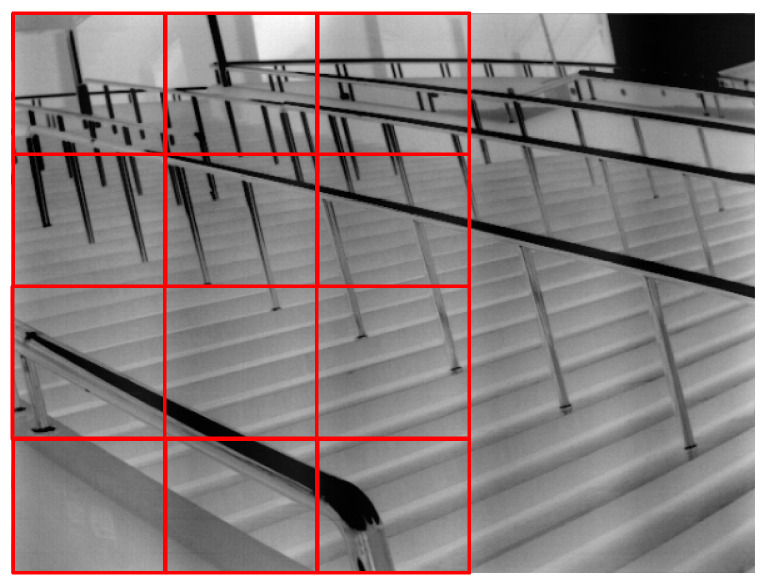
The process of image block segmentation.

**Figure 6 sensors-23-08716-f006:**
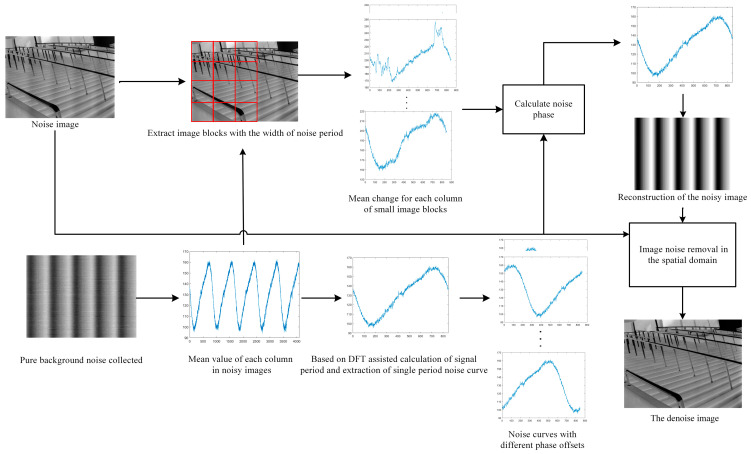
Flowchart of the proposed denoising algorithm. The flowchart diagram illustrates that the first step involves employing Fourier Transform to calculate the periodicity of the absorbed signals. The upper section of the diagram shows the extraction of a single-period signal from images containing objects, represented as a one-dimensional curve. Meanwhile, the lower section represents the extraction of single-period signals with different phases using a black body. Subsequently, noise reduction is achieved in the spatial domain through the utilization of correlation.

**Figure 7 sensors-23-08716-f007:**
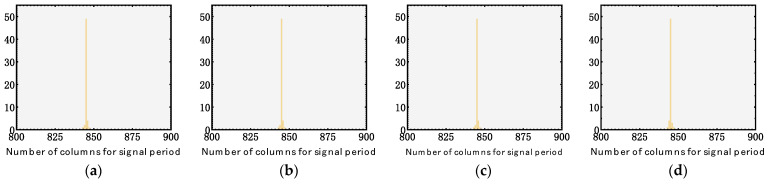
Histograms for noise periods against cycle counts captured in 0 (**a**), 6 (**b**), 12 (**c**), and 18 (**d**) hour(s) after the initial power-on of an infrared linear array detector. In the histogram, the yellow bars represent the counts of signals at different periods.

**Figure 8 sensors-23-08716-f008:**
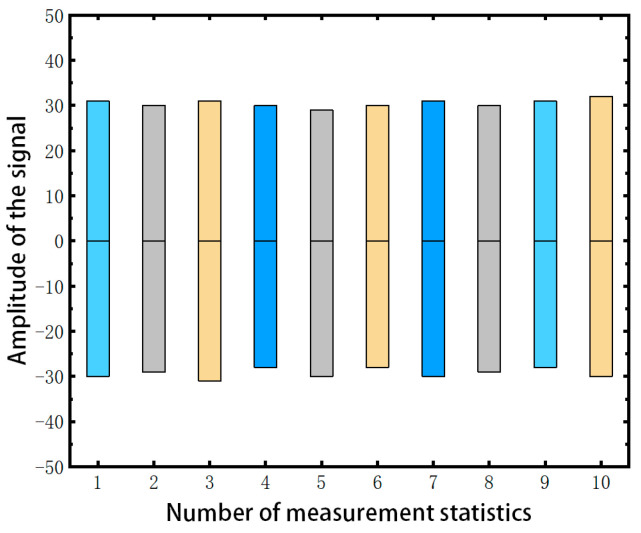
Noise amplitude fluctuation. It shows the amplitude fluctuations of the signal typically range between approximately −30 and 30.

**Figure 9 sensors-23-08716-f009:**
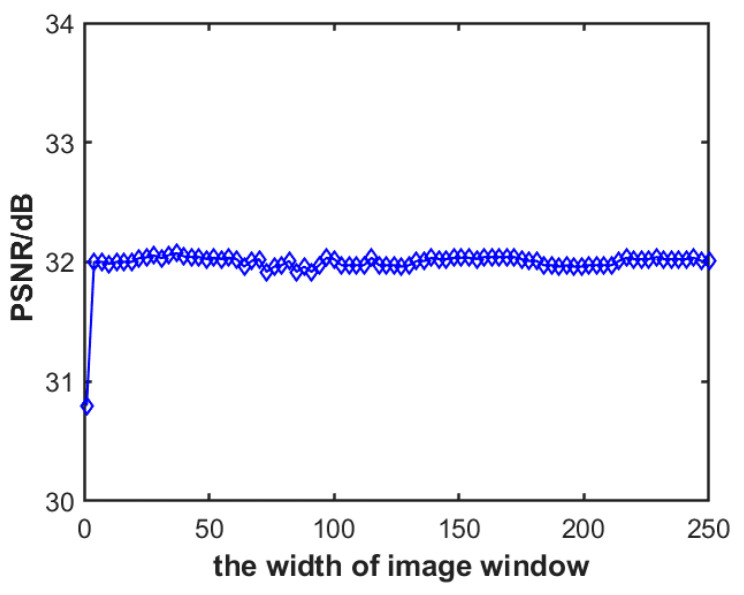
The influence of image block width on algorithm denoising.

**Figure 10 sensors-23-08716-f010:**
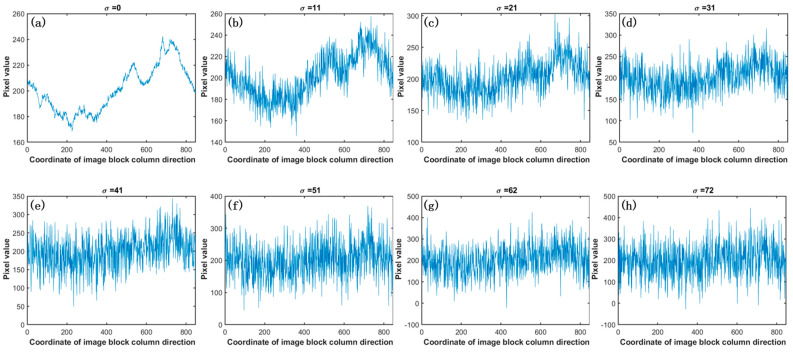
Simulating image background complexity using Gaussian noise with zero mean and diverse variances. With an increase in the variance of added noise, there is a noticeable amplification in the signal curve’s fluctuations within the images. This trend leads to a progressive loss of clarity in the distinctive features of the original signal, resulting in an overall blurring effect.

**Figure 11 sensors-23-08716-f011:**
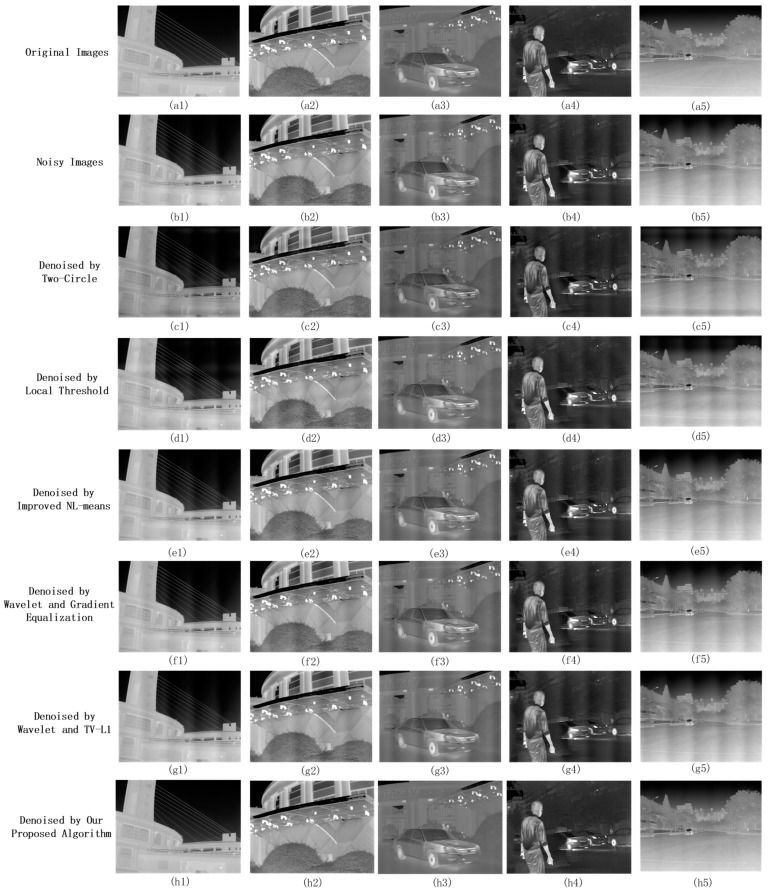
Simulate image denoising effect: (**a**) original images, (**b**) noisy images; (**c**) Two-Circle; (**d**) Local Threshold; (**e**) improved NL-means; (**f**) Wavelet Gradient-Equalization; (**g**) Wavelet TV-L1; (**h**) our proposed algorithm.

**Figure 12 sensors-23-08716-f012:**
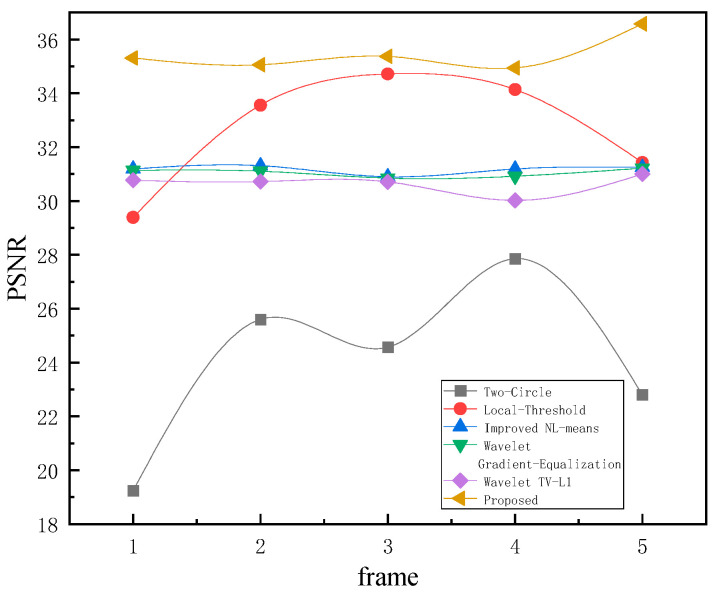
Line chart for Table 2: denoising results of different algorithms.

**Figure 13 sensors-23-08716-f013:**
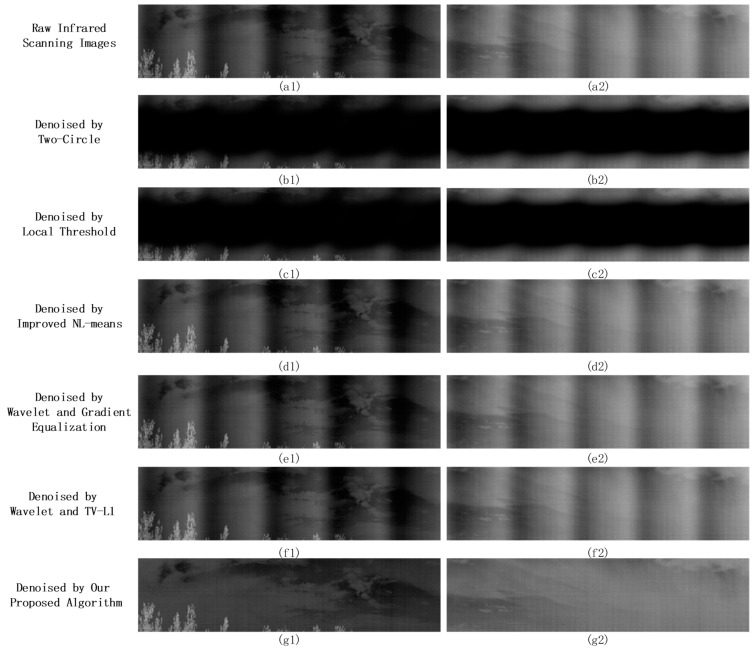
Denoising effect comparison on real infrared images. Both “Denoised by Two-Circle” and “Denoised by Local Threshold” methods, when applied in the frequency domain, led to a significant deterioration of the low-frequency information in the images. Conversely, “Denoised by Improved NL-means”, “Denoised by Wavelet and Gradient Equalization”, and “Denoised by Wavelet and TV-L1” methods exhibited limitations in effectively eliminating this low-frequency periodic noise. In contrast, our proposed algorithm demonstrated a notable capability to successfully remove this type of noise, providing a more effective denoising solution.

**Table 1 sensors-23-08716-t001:** Effect of Gaussian noise on algorithmic denoising.

Standard Deviation σ	PSNR
0	32.83
7.65	32.72
17.85	32.71
28.05	32.72
38.25	32.70
48.45	32.71
58.65	32.30
68.85	31.69
79.05	31.65

**Table 2 sensors-23-08716-t002:** Denoising results of different algorithms.

Image	PSNR					
	Two-Circle	Local Threshold	Improved NL-Means	Wavelet Gradient-Equalization	Wavelet TV-L1	Proposed
IM1	19.24	29.39	31.19	31.14	30.77	35.31
IM2	25.60	33.56	31.31	31.11	30.72	35.06
IM3	24.57	34.72	30.90	30.85	30.71	35.37
IM4	27.86	34.14	31.19	30.92	30.02	34.94
IM5	22.80	29.43	31.25	31.22	31.00	36.58
Dataset	26.0	31.34	31.00	30.84	30.04	34.11

**Table 3 sensors-23-08716-t003:** SSIM index of different algorithms.

Image	SSIM					
	Two-Circle	Local Threshold	Improved NL-Means	Wavelet Gradient-Equalization	Wavelet TV-L1	Proposed
IM1	0.8319	0.9004	0.9213	0.9192	0.9090	0.9880
IM2	0.9441	0.9701	0.9740	0.9679	0.9496	0.9889
IM3	0.9813	0.9917	0.9872	0.9855	0.9789	0.9963
IM4	0.9793	0.9908	0.9841	0.9730	0.9369	0.9943
IM5	0.9175	0.9383	0.9347	0.9323	0.9173	0.9663
Dataset	0.9345	0.9596	0.9632	0.9566	0.9374	0.9874

**Table 4 sensors-23-08716-t004:** Average run time of different algorithms.

Algorithm	Time Taken (s)				
(Image Resolution)	1024 × 4096	1024 × 6144	1024 × 8192	1024 × 10,240	1024 × 12,288
Two-Circle	0.26	0.46	0.59	0.74	0.91
Local Threshold	0.21	0.34	0.46	0.59	0.75
Improved NL-means	310.24	508.25	665.03	850.93	1006.72
Wavelet and Gradient-Equalization	50.27	80.78	99.41	128.73	147.27
Wavelet and TV-L1	9.63	16.15	21.60	27.50	33.51
Ours (FACD)	0.07	0.11	0.14	0.18	0.22

## Data Availability

Not applicable.

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
