# Peer review of "Overcoming Periodic Stripe Noise in Infrared Linear Array Images: The Fourier-Assisted Correlative Denoising Method"

_sensors, 2023, doi:10.3390/s23218716_

Round 1
Reviewer 1 Report
This is an article with engineering value, which is used to remove stripe noise. I believe this article have some problems:
1. In Introduction, the deep learning-based method should be introduced more. Recently, the self-supervision denoising is popular, because thry can remove noise without training image, such as "Noise2Noise: Learning Image Restoration without Clean Data" and "Fractional Variation Network for THz Spectrum Denoising without Clean Data"
2. Table 2 only provides PSNR, and more evaluation methods need to be applied., suah as SSIM.
3. In 3.6, the authors provide runtime, and I prefer to see the algorithm's runtime change as the image size increases. The subsampling operator can be used to change the image resolution for further evaluation
Moderate editing of English language required
Reviewer 2 Report
Minor Revision
This research offers a novel solution to the prevalent challenge of low-frequency periodic signal noise, induced by electromechanical interferences in infrared linear-array detectors. A denoising algorithm, underpinned by signal correlation principles, has been introduced. This research offers a novel solution to the prevalent challenge of low-frequency periodic signal noise, induced by electromechanical interferences in infrared linear-array detectors. A denoising algorithm, underpinned by signal correlation principles, has been introduced., from my viewpoint, if following issues were solved, it can be published in this famous, reputation journal.
1. The images in Figure 1 and Figure 4 do not have labels (a, b) to distinguish between different parts, and the image titles are not consistent with the formats of other images.
2. The text sizes in Figure 3 and Figures 6, 7, and 8 are inconsistent.
3. What is the framework of the denoising algorithm based on, and specifically introduce what the principle of FACD is.
4. In Table 2, it is recommended to add a line chart to represent data, which is more intuitive.
5. Please revise the conclusion in paragraphs. Conclusions are not just about summarising the key results of the study, it should highlights the insights and the applicability of your findings/results for further work. Please make it more concise and show only the high impact outcomes.
Round 2
Reviewer 1 Report
After modification, the quality of the article has been greatly improved, and it is recommended to accept
After modification, the quality of the article has been greatly improved, and it is recommended to accept